# Analysis of the Potential of an Increase in Yeast Output Resulting from the Application of Additional Process Wastewater in the Evaporator Station

**Barbara Włodarczyk** and **Paweł P. Włodarczyk** *

Institute of Technical Science, Faculty of Natural Sciences and Technology, University of Opole,
Dmowskiego str. 7-9, 45-365 Opole, Poland; barbara.wlodarczyk@uni.opole.pl
* Correspondence: pawel.wlodarczyk@uni.opole.pl; Tel.: +48-077-401-6717

**Abstract:** This paper reports the results of an analysis of process wastewater streams in the context of an increase in yeast production. This research is based on the analysis of data from the biggest yeast factory in Europe. The research presented in this paper involves the analysis of the influence of direction of additional wastewater into the evaporator station on yeast production. In the process wastewater, nitrogen is mainly present in organic forms. The analysis reported in this paper involves the concentration of total nitrogen in wastewater streams, as it is the main parameter applied to determine the amount of wastewater that can be applied in agricultural fields. Directing additional wastewater into the evaporator station can offer a simultaneous increase in the volume of its use in the field of agriculture and will ultimately yield an increase in productivity (under conditions where additional pressure on the natural environment is not exerted). The results obtained in this analysis were an increase in production of $\eta_{Yp} = 0.1027$, corresponding to about 6500 Mg of yeast per year. This is a feasible value, which can be derived from the existing agricultural field area and the properties of the evaporator station in the factory. At the same time, the same increase in the volume of organic fertilizer is obtained. This fertilizer is generated as a byproduct of the pre-treatment of wastewater at the evaporator station. Thus, the increase in the production of the fertilizer can have a positive effect on fields in local farms, which are typically the recipients of this fertilizer.

**Keywords:** environmental engineering; production increase; yeast wastewater; sustainability

## 1. Introduction

The food industry forms an important sector of the European economy [1]. It is involved in processes related to the increase in gross domestic product as well as satisfying domestic demand. It also contributes to the creation of new jobs and plays an important role in the international trade exchange. Consequently, we can state that the role and relevance of the food industry are huge. However, the production potential of the food industry and its capabilities are influenced by the potential held in the natural resources and technical measures that are accessible in a given country. These factors, directly or indirectly, affect the production of food products. However, when the importance of the food industry in the national economy and individual countries of the European Union is investigated, the impact of agriculture on the natural environment should be particularly taken into account. Measures aimed at reducing the negative impact of human activity on the environment are increasingly being taken into account in production [1,2]. In accordance with the idea of sustainable development, the rational use of natural resources will meet the needs of the present and future generations. In addition, the measures applied to limit the generation of large amounts of waste or process wastewater can, consequently, contribute to a smaller financial burden for enterprises associated with their disposal [2].

In recent years, the Polish food sector has undergone tremendous changes, both in terms of technology and measures linked to environmental protection. Thanks to the constant technical, engineering, and innovative development, Poland has become one of the leading food producers in Europe. One of the factories belonging to the group of European leaders, in terms of the variety of products on offer, is provided by the yeast factory analyzed in this paper [3,4]. The yeast industry consumes significant amounts of water for the production of yeast and, consequently, huge amounts of process wastewater (technological wastewater) are produced, which need to be later utilized. Process wastewater from the yeast industry mainly originates from decanting and wort thickening. This wastewater is of plant origin and due to the high content of nitrogen, potassium, and organic substances, it can be (and often is) deposited on agricultural fields (AF) to play the role of a fertilizer. Chemical analysis of the level of humus in soils, in which process wastewater from the yeast plant is deposited, does not demonstrate an increase in the accumulation of heavy metals. The values of concentrations of trace elements in soils irrigated with yeast wastes are much lower than the levels that are permitted in soils and plants. The parameters of natural geochemical values in soils are, thus, not exceeded [5]. Research has also been carried out into the treatment of yeast wastewater by application of nano-filtration, electro-coagulation, or in the use of technological yeast wastes for application in the direct generation of electricity using microbial fuel cells [6–12]. The deposition of process wastewater derived from the yeast industry on agricultural fields (AF) seems to represent the most rational solution [5,13,14]. Cultivation on AF fertilized with yeast technological wastewater is characterized by a considerable efficiency. In addition, such wastewater contains organic substances that have a positive effect on the fertilized soils and contributes to the improvement of soil quality in the vicinity of the plant [15]. However, it is possible to irrigate AF using only limited amounts of process wastewater [15,16]. The main reason that the volume of wastewater directed to AF needs to be limited is associated with the concentration of nitrogen in this wastewater [15,17]. For this reason, wastewater is often pre-treated in an evaporator station (ES), in order to decrease the nitrogen concentration in wastewater that is applied to AF. Theoretically, when a larger amount of wastewater is routed to process in an ES, the possibility of using larger amounts of process wastewater in AF is feasible, which, in turn, can lead to an increase in the yeast output [18]. Previous studies [18,19] have shown the necessity of a detailed analysis of the wastewater treatment system in yeast factories. The analyzed factory includes a three-stage wastewater treatment system, comprised of an agricultural field (AF), an evaporator station (ES), and a biological wastewater treatment plant (WWTP). The analysis of the three-stage wastewater treatment system (AF–ES–WWTP) in a yeast factory will provide the development of a solution leading to an increase in production, without the plant exerting additional pressure on the natural environment. A detailed analysis of technological wastewater streams will determine the level in which the production of yeasts (and fertilizers forming a by-product) can be increased in the ES. The use of the ES is essential, in order to reduce the concentration of nitrogen in process wastewater and the possibility of increasing the amount of their deposition on AF, in particular in the case of large-scale production.

As a result of irrigation, when applying process wastewater derived from yeast production, large amounts of total nitrogen are introduced into the soil, mainly in organic form. Nitrogen transformations occurring in the soil primarily includes decomposition processes involving the mineralization of organic nitrogen, nitrification, and de-nitrification, consequently leading to the formation of gaseous nitrogen molecules. Secondly, there are synthetic processes leading to the formation of organic compounds [20,21]. This process involves mineral nitrogen uptake by plants and an immobilization process involving the combination of mineral nitrogen into the biomass of soil micro-organisms and then into humus compounds [20,22].

The process wastewater from yeast production forms a completely safe substance, in terms of sanitary characteristics. In addition, the concentrations of heavy metals are at permissible levels by the law. Therefore, this wastewater can be employed directly for crop and vegetable fertilization. In particular, plants with a high demand for nitrogen and potassium, such as beets, rapeseed, or corn, can provide a preferred use. If there is no need for the production of fodder crops in a specific area,

long-term irrigation with process wastewater derived from yeast production offers a suitable source of water to be applied in industrial plants and cereals [23,24]. Therefore, irrigation applying process wastewater from yeast production to agricultural fields provides the most rational method of its treatment. However, irrigation by application of process wastewater is limited by the regulations guiding the amount of substances introduced into soils and waters [25,26].

This analysis of wastewater streams is based on data obtained from the Lesaffre yeast factory (Wołczyn, Poland). Due to the high quality of the final products and agricultural use of process wastewater generated in the factory, Lesaffre Poland follows the guidelines of sustainable development in its environmental protection program. The factory taken for the analysis is the largest yeast production factory in Europe, and thus offers a representative object for the present analysis. The purpose of this paper involves the determination of the theoretical and feasible value of production growth, based on variation in the amount of technological wastewater designed for application in ES.

## 2. Analysis of the Potential for the Increase in Yeast Production and Results

The research presented in this paper involves the analysis of the influence of the direction of additional wastewater into the ES on an increase in yeast production. The analysis reported in this paper involves the concentration of total nitrogen in wastewater streams ($S_i$), as it is the main parameter applied to determine the amount of wastewater that can be applied in AF. In addition, this analysis accounts for the use of the AF–ES–WWTP system. The increase in yeast and fertilizer production was achieved by assuming a constant area of AF. Nitrogen is a basic parameter in limiting the amount of wastewater directed to AF. Moreover, the analysis includes also the AF–ES–WWTP.

Figure 1 shows the scheme of the yeast production process and the AF–ES–WWTP system. Figure 1A shows the sources of process wastewater in the yeast factory, and Figure 1B shows the paths of the wastewater streams. The $S_6$ stream of process wastewater (Figure 1A) is directed directly to the WWTP (Figure 1B). This stream is not directed to the AF, because it is a domestic wastewater resulting from the washing of the devices. This stream of the wastewater includes detergents, oils, domestic pollutants, chemicals, heavy metals, metal, and rubber filings, among other waste products. However, after treatment of this wastewater (as clean wastewater), it can be directed to the wastewater pumps, in order to dilute the process wastewater stream $S_9$ directed to the AF. The concentration of nitrogen ($C_1$) in the $S_1$ stream is very high, but the $S_1$ stream has low volume (see Table 1). The concentration of the nitrogen ($C_8$) in the $S_8$ stream is high, but the $S_8$ stream is negligible. This stream is needed only for microbes for the operation of the WWTP (Figure 1B). Thus, these streams ($S_1$ and $S_8$) do not have a large effect on the final nitrogen concentration (of the $S_9$ stream).

Process wastewater ($S_2$ and $S_3$) from first and second centrifuges is characterized by the highest concentrations of pollutants. Although the wastewater from clarification of molasses ($S_1$) is characterized by even higher pollutant concentrations, the volume of this wastewater is very small. The volume of the process wastewater and amount of nitrogen in process wastewater are presented in Table 1.

**Table 1.** Volume and nitrogen of the process wastewater.

| Streams of Process Wastewater | | $S_1$ | $S_2$ | $S_3$ | $S_4$ | $S_5$ |
|---|---|---|---|---|---|---|
| | | wort decantation | 1st centrifuge separator | 2nd centrifuge separator | 3rd centrifuge separator | vacuum filters |
| volume | [m$^3$/yr] | 400 | 17,150 | 232,850 | 53,300 |
| $C_i$ of total N | [kg/m$^3$] | 8.5 | 3.3 | 2.1 | 0.65 |

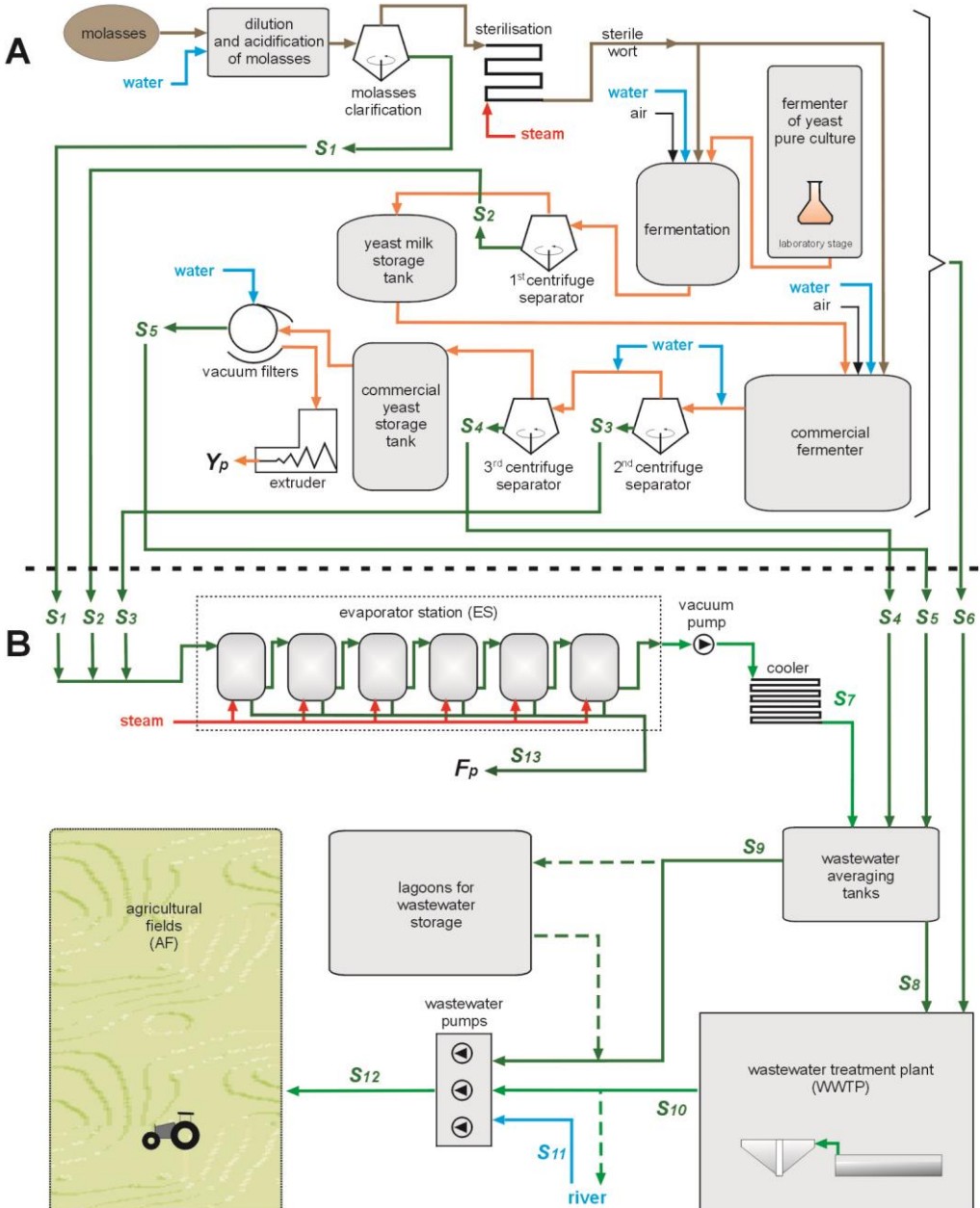

**Figure 1.** Scheme of the wastewater sources (**A**) and the three-stage wastewater treatment system (**B**) in the analyzed yeast factory. $Y_p$, yeast production; $F_p$, fertilizer production.

$S_i$, stream of wastewater [m³/year];

$S_1$, stream of process wastewater after molasses clarification;

$S_2$, stream of process wastewater from the first centrifuge separator;

$S_3$, stream of process wastewater from the second centrifuge separator;

$S_4$, stream of process wastewater from the third centrifuge separator;

$S_5$, stream of process wastewater from the vacuum filters;

$S_6$, stream of domestic wastewater and after cleaning of the process devices;

$S_7$, stream of pre-treated process wastewater after the evaporator station (ES);

$S_8$, little stream of process wastewater directed to the wastewater treatment plant (WWTP);

$S_9$, stream of process wastewater directed to the agricultural fields (AF) from equalized wastewater;

$S_{10}$, stream of cleaned wastewater directed to the AF;

$S_{11}$, stream of water from the river;

$S_{12}$, the $S_9$ stream of process wastewater (directed to AF) diluted by $S_{10}$ or $S_{11}$ stream;

$S_{13}$, stream of pollutants from the ES ($F_p$ fertilizer).

The stream $S_9$ [m³/yr] of process wastewater (Figure 1B) directed to the AF can be described as Equation (1)

$$S_9 = S_1 + S_4 + S_5 + S_7 - S_8. \tag{1}$$

The $S_9$ stream (Figure 1B) is averaged in the tanks of wastewater equalization. The nitrogen concentration (the analysis is based on nitrogen concentration) in the $S_9$ stream can be described by Equation (1a):

$$S_9 C_9 = S_1 C_1 + S_4 C_4 + S_5 C_5 + S_7 C_7 - S_8 C_8, \tag{2}$$

where $C_i$ is the concentration of nitrogen in the stream $S_i$ [kg/m³].

The stream ($S_8$) that is routed to WWTP is very small; however, it is needed for the adequate performance of micro-organisms in the WWTP. For this reason, it was neglected in the computations. Hence, the equation in 1a assumes the form

$$S_9 C_9 = S_1 C_1 + S_4 C_4 + S_5 C_5 + S_7 C_7. \tag{3}$$

The concentration of pollutants in the $S_9$ stream can be described as

$$S_9 = \frac{S_1 C_1 + S_4 C_4 + S_5 C_5 + S_7 C_7}{C_9}, \tag{4}$$

assuming that (taking into account that the $S_9$ stream is omitted)

$$S_9 = S_1 + S_4 + S_5 + S_7. \tag{5}$$

The stream $S_{12}$ is the stream $S_9$ diluted by the $S_{10}$ or $S_{11}$ stream. Therefore, the amount of nitrogen in the $S_{12}$ stream (directed to the AF) is equal to the amount of nitrogen in the $S_9$ stream.

The amount of nitrogen (deposited into the AF) is limited by the Regulation of the Minister of Environment of 16 November 2014 (on the conditions to be met when introducing wastewater into waters or into the ground, and on substances particularly harmful to the aquatic environment) and by the Directive 2010/75/EU of 24 November 2010 on the industrial emissions (integrated pollution prevention and control) [25,26]. The analyzed yeast factory meets all Polish and EU standards for the amount of nitrogen deposited into the AF. Due to the high nitrogen pollution of the soils and the waters, an increase in yeast production is possible only if all the standards are met [24–26].

However, within these regulations, it is possible to control this parameter according to, for example, the kinds of plants grown on the AF. The $D_{WW}$ dose (of the process wastewater) that can be utilized on the one hectare of AF (depending on the $C_N$ nitrogen concentration) can be determined from Equation (6) [15,16]

$$D_{WW} = \frac{p_N \cdot k_{dN}}{C_N \cdot k_{uN} \cdot E_N}, \tag{6}$$

where

$D_{WW}$ is the maximum dose of the process wastewater which can be directed to the AF [m³/ha·yr];

$p_N$ is the total nitrogen demand for the plants which are cultivated on the AF [kg/ha·yr];

$k_{dN}$ is the total nitrogen demand factor (if the AF are fertilized by only the process wastewater, $f_N$ = 1);

$C_N$ is the total nitrogen concentration in the process wastewater directed to the AF [kg/m³];

$k_{uN}$ is the total nitrogen utilization factor (depending on the irrigation system); and

$E_N$ is the nitrogen fertilizer equivalent (for process wastewater from yeast production, $E_N$ = 1).

The volume of nitrogen, $m_{NAF}$, which can be deposited on the AF, can be defined as

$$m_{NAF} = D_{WW} \cdot C_9, \tag{7}$$



where

$m_{NAF}$ is the amount of nitrogen which can be deposited into the AF [kg/ha·yr], and
$C_9$ is the concentration of nitrogen in stream $S_9$ [kg/m$^3$].
The yeast production volume, $Y_p$, is limited by the amount of nitrogen [Mg/yr]

$$Y_p = \frac{A_{AF} \cdot m_{NAF}}{m_{NpY}}, \tag{8}$$

where

$A_{AF}$ is the area of the AF [*ha*], and $m_{NpY}$ is the amount of nitrogen in the process wastewater stream (directed to the AF) per 1 Mg of yeast production [kg/Mg].
Thus, the area of the AF $A_{AF}$ [ha] can be described as

$$A_{AF} = \frac{m_{NpY}}{m_{NAF}} \cdot Y_p. \tag{9}$$

The nitrogen demand of plants is relative to the type of agricultural plants produced in the AF.
The maximum amount of the process wastewater $D_{WWr}$ that is routed to AF can be expressed by the relation

$$D_{WWr} = \sum a_i D_{WWi}, \tag{10}$$

where

$a_i$ is the share of the individual plant crops $i$ in the AF; $D_{WWi}$ is the maximum dose of process wastewater for plants that are cultivated on the AF [m$^3$/ha·yr]; and $n$ is the number of crops,

where

$$\sum_{i}^{n} a_i = 1. \tag{11}$$

The volume of process wastewater $V_{ww}$ that is applied annually to the AF can be determined on the basis of Equation (12)

$$V_{WW} = D_{WWr} \cdot A_{AF,}. \tag{12}$$

When we take into account Equations (7) and (12), the volume of yeast production $Y_p$ (Equation (8)) can be expressed in the form

$$Y_p = \frac{V_{WW} \cdot C_9}{m_{NpY}}. \tag{13}$$

For the specific surface area of the agricultural land (in the investigated case, with regard to the agricultural land irrigated by the analyzed plant), the values of the parameters $p_N$, $k_{dN}$, $k_{uN}$, $E_N$, and $A_{AF}$ assume constant values. Hence, the value $D_{wwr}$ is also constant. For this reason, it is possible to apply an indicator $W_N$ [kg/year] to determine the amount of total nitrogen that is applied to the agricultural fields of the analyzed yeast plant. This parameter can be expressed in the following form:

$$W_N = \frac{p_N \cdot k_{dN} \cdot A_{AF}}{k_{uN} \cdot E_N}. \tag{14}$$

Therefore, Equation (12) can be described as

$$V_{WW} = \frac{W_N}{C_9}. \tag{15}$$

Using the indicator $W_N$ allows us to write Equation (12) as

$$Y_p = \frac{W_N}{m_{NpY}}. \tag{16}$$

It should be noted that the parameter $m_{NpY}$ is determined only by the technology of yeast production. This parameter is specific for each specific yeast factory. However, directing a larger amount of process wastewater to the ES will result in a lowering of the nitrogen concentration $C_9$ in the wastewater stream $S_9$ (directed to the AF). Therefore, lowering the concentration $C_9$ in the stream $S_9$ will result in decrease in the value of the $m_{NpY}$ parameter. Thus, the value of $Y_p$ (Equation (16)) will be higher.

Currently, only the process wastewater coming out coming out from the three separators (streams $S_1$, $S_2$, and $S_3$) is sent to the ES. Nevertheless, directing a certain part of stream $S_4$ to the ES will enable a further reduction of the nitrogen (in reality, this affects all pollutants, but this analysis is based on the nitrogen concentration in the wastewater streams directed to the AF) concentration and, in consequence, increases both the maximum volume of the process wastes distributed in the AF (per year) and the yeast production.

Figure 2 shows the AF–ES–WWTP system with the paths of the wastewater streams and an additional $nS_4$ stream, directed to the ES.

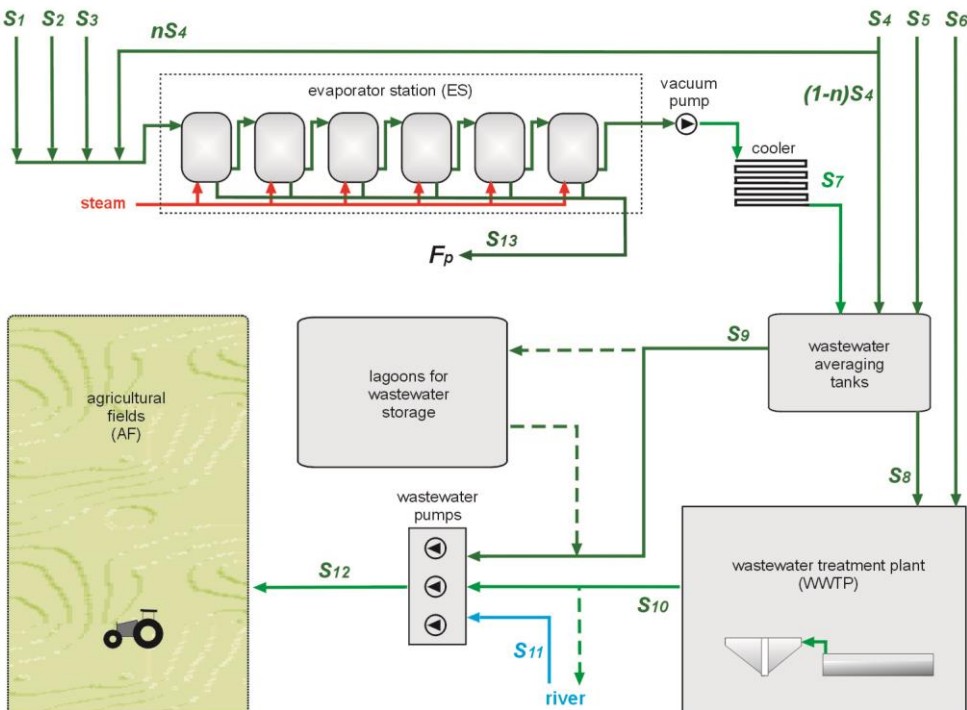

**Figure 2.** Scheme of three-stage wastewater treatment (agricultural field, evaporator station, and a biological wastewater treatment plant) system, with an additional $nS_4$ stream directed to the evaporator station.

The streams $S_1$ and $S_8$ are negligibly small. Thus, they do not affect the final concentration of nitrogen in the stream, expressed as $S_9$. Therefore, these streams can be disregarded in the calculations. Thus, the part of stream $S_4$ of wastewater directed to the ES (Figure 2) can be described as

$$nS_4 = S_4 - (1-n)S_4, \tag{17}$$

where

$nS_4$ is the part of the $S_4$ wastewater stream directed to the ES, $(1-n)S_4$ is the part of the $S_4$ process wastewater which joins the $S_9$ stream (directed to the AF), and n ∈ (0, 1).

The $S_9$ stream of process wastewater [m$^3$/yr] (Figure 2), which is directed to the AF, can be described (based on Equations (5) and (17)) as

$$S_9 = (1 - n)S_4 + S_5 + S_7. \tag{18}$$

The nitrogen concentration in the $S_9$ stream (after a part $n$ of the $S_4$ stream is directed to the ES) is described (based on Equations (3) and (18)) by

$$S_9 C_9 = (1 - n)S_4 C_4 + S_5 C_5 + S_7 C_7. \tag{19}$$

Taking into account Equations (4) and (19) (and omitting the $S_1$ stream), the $C_9$ nitrogen concentration, after directing the $n$ part of the $S_4$ stream to the ES, can be described as

$$C_9 = \frac{(1 - n)S_4 C_4 + S_5 C_5 + S_7 C_7}{S_9}. \tag{20}$$

The first part of this paper involved the determination of the theoretical increase in yeast production, on the basis of an assumption regarding complete nitrogen neutralization taking place in ES. This means that concentration expressed as $C_7$ is equal to 0. Thus, Equation (20) can be written in the form

$$C_9 = \frac{(1 - n)S_4 C_4 + S_5 C_5}{S_9}. \tag{21}$$

The mass of nitrogen $m_{NAF}$, which is directed annually to the agricultural fields (AF), can be described as

$$m_{NAF} = (1 - n)S_4 C_4 + S_5 C_5. \tag{22}$$

It is possible to calculate the amount of nitrogen in the process wastewater stream (directed to the AF) per 1 Mg of the yeast production during the actual yeast production:

$$m_{NpY} = \frac{S_4 C_4 + S_5 C_5}{Y_p}. \tag{23}$$

Taking into account Equations (22) and (23), we can calculate the total amount of nitrogen in the process wastewater stream ($m_{NpYplus}$) per 1 Mg of yeast production, during yeast production, after directing the $nS_4$ stream to the ES:

$$m_{NpYplus} = \frac{(1 - n)S_4 C_4 + S_5 C_5}{Y_{pplus}}, \tag{24}$$

where

$m_{NpYplus}$ is the total amount of nitrogen in the process wastewater stream (directed to the AF) per 1 Mg of yeast production, after directing the $nS_4$ stream to the ES [kg/Mg]; and $Y_{pplus}$ is the yeast production volume, after directing the $nS_4$ stream to the ES [Mg/yr].

Based on Equations (16) and (22)–(24), it can be assumed that

$$m_{NAF} = m_{NpY} \cdot Y_p = m_{NpYplus} \cdot Y_{pplus}. \tag{25}$$

Therefore,

$$\frac{Y_{pplus}}{Y_p} = \frac{m_{NpY}}{m_{NpYplus}}. \tag{26}$$

Taking into account Equations (23) and (24), the ratio of production after directing the $nS_4$ stream to the ES to the current production, can be described as

$$\frac{Y_{pplus}}{Y_p} = \frac{S_4 C_4 + S_5 C_5}{(1 - n)S_4 C_4 + S_5 C_5}. \tag{27}$$

The increase in yeast production ($\eta_{Yp}$) after directing the $nS_4$ stream to the ES can be described as

$$\eta_{Yp} = 1 - \frac{Y_{pplus}}{Y_p}. \tag{28}$$

Therefore,

$$\eta_{Yp} = \frac{nS_4C_4}{(1-n)S_4C_4 + S_5C_5}. \tag{29}$$

In addition, along with an increase in yeast production ($\eta_{Yp}$) after the stream $nS_4$ is routed to ES, at the same time, an additional volume of pollutants (mainly nitrogen, but also potassium or organic substances) is derived as the organic fertilizer $F_p$ (Figures 1B and 2—$S_{13}$). The increase in the amount of fertilizer $\eta_{Fp}$ is relative to the additional volume $nS_4$ of the process wastewater that is routed to the ES, and is directly proportional to the increase in yeast production $\eta_{Yp}$.

The $F_p$ organic fertilizer obtained in the ES has a dark brown color and a liquid-honey consistency. This fertilizer meets all standards for substances entering the soil [27,28]. Moreover, this product is readily used by the local farmers as an organic fertilizer and as an animal feed additive [28,29]. This fertilizer is introduced into the soil mainly during plowing.

The increase in production of both $Y_p$ and $F_p$ was analyzed, assuming a constant surface area of the AF, as well as constant crops. In the yeast factory adopted in this analysis, the area of AF is equal to 1800 ha. Industrial and cereal crops (in the ratio of 3:2) are cultivated in these AF.

On this basis, the possible increase in yeast and fertilizer production could be estimated on the basis of the relation in (29).

The three-stage wastewater treatment system implemented in the plant leads to a complete neutralization of the pollutants contained in the process wastewater. However, a single-stage wastewater treatment system (e.g., just the ES) is not capable of completely eliminating the existing concentration of pollutants. In the conditions of the actual production process (derived on the basis of data from the factory), it is necessary to take into account the actual ratio of nitrogen neutralization at the ES. Taking into account the data summarized in Tables 2 and 3, the calculations offer a curve representing the increase in production for the actual production facility (i.e., in the analyzed yeast factory).

This study assumed that:

- Nitrogen concentration is reduced to 0% at the ES (theoretical calculation);
- Nitrogen concentration is reduced by 52% at the ES (for real parameter of ES, Table 2); and
- The nitrogen demands for the industrial and cereal plants grown in the AF (Table 3) is accounted for in the Equations (6) and (14).

**Table 2.** Percentage of total nitrogen reduction at individual stages of the three-stage wastewater treatment (AF–ES–WWTP) system.

| Reduction of N [%] | | |
|---|---|---|
| WWTP | AF | ES |
| 19 | 29 | 52 |

**Table 3.** Data for plants cultivated in the AF.

| Parameter | Value |
|---|---|
| $p_N$: total nitrogen demand for plants cultivated on the AF. | 300 [kg/ha·yr] |
| $k_{dN}$: total nitrogen demand factor (if AF are fertilized by only the process wastewater). | 1.00 |
| $k_{uN}$: total nitrogen utilization factor (depending on the irrigation system); in analyzed factory, sprinkling irrigation. | 0.95 |
| $E_N$: total nitrogen fertilizer equivalent. | 1.00 |

Figure 3 presents the curve representing the increase in the yeast output $\eta_{Yp}$ fertilizer production $\eta_{Fp}$ as a function of $n$ (the ratio of the flux $S_4$ that is routed into the ES).

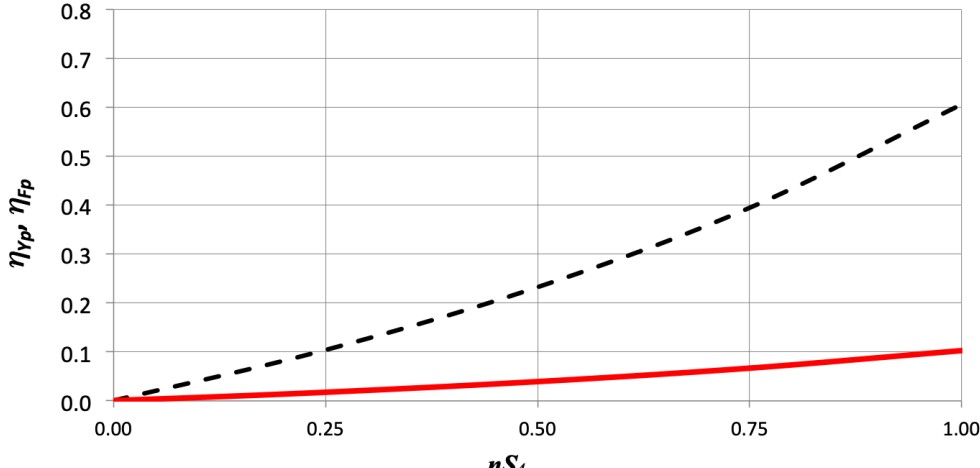

**Figure 3.** The increase in yeast production $\eta_{Yp}$ and fertilizer production $\eta_{Fp}$ versus the volume fraction $n$ (the part of stream $S_4$ directed to the ES). The dotted line represents the theoretical increase, and the red line is the real increase possible to achieve, considering the data from the analyzed factory (at current efficiency of the ES).

As we can see from the top curve (Figure 3, black dotted line), when the entire volume of the wastewater $S_4$ (n = 1) is used as an input to the ES, we note an increase in the yeast and fertilizer output by just over 60% ($\eta_{Yp}$ = 0.6057). This provides a considerable increase; however, this is only theoretical, since a complete neutralization of total nitrogen is assumed. Taking into account the efficiency of the existing evaporator station (Tables 1–3), the actual achievable increase in production in the analyzed plant (Figure 3, red line) was determined to be just over 10% ($\eta_{Yp}$ = 0.1027).

## 3. Discussion and Conclusions

The analysis conducted for the purposes of the present study demonstrates that it is possible to increase yeast production ($\eta_{Yp}$) by directing an additional stream of process wastewater ($nS_4$) into the ES. This increase can be achieved by the application of the current area of agricultural fields ($A_{AF}$), which are located on the site of the yeast plant. At the same time (like $\eta_{Yp}$), the same increase in the volume of organic fertilizer $\eta_{Fp}$ can be achieved. This fertilizer ($F_p$) is generated as a byproduct of the pre-treatment of wastewater at the ES.

Theoretically, it is possible to obtain a significant increase in yeast production $\eta_{Yp}$ = 0.6057 (Figure 3, black line). However, if this level should be assumed, a complete neutralization of nitrogen in the ES needs to be achieved (theoretically increasing yeast production). Using the parameters of the actual reduction of nitrogen concentration in the ES of the analyzed factory, the maximum production increase was reached $\eta_{Yp}$ = 0.1027 (Figure 3, red line). This is the maximum increase in production obtained for $n$ = 1 (i.e., where $S_4$ is completely directed into the ES). In the analyzed plant, 6.46 m$^3$ of wastewater is generated per 1 Mg of yeast output (data from the factory). If we know the amount of process wastewater generated during actual production (Table 1), we can determine the annual production $Y_p$, amounting to about 64,000 Mg. The increase in production ($\eta_{Yp}$) at just over 0.1 corresponds to about 6500 Mg of yeast per year. This is the feasible value that can be derived from the existing area of AF and the ES located at the factory. In this case, there is no need to intervene in the existing AF–ES–WWTP system; thus, this is a beneficial solution for the operation of the yeast factory. As a result of the increase in the amount of process wastewater applied to the evaporator station ($nS_4$), the ES is faced with the need to apply greater volumes of energy to operate. However, the analyzed factory includes a boiler burning biomass. In this way, a reduction in the operating costs of the evaporator station is feasible.

In addition, the cost of using an ES can be compensated as a consequence of the increase in revenues resulting from the increase in yeast and fertilizer production ($\eta_{Yp}$ and $\eta_{Fp}$).

In addition, the increase in fertilizer production ($\eta_{Fp}$) will have a positive effect, not only on the productivity of the AF at the plant, but also on the fields of individual farmers locally. This fertilizer ($F_p$) can be used in large quantities to provide a significant enrichment of soils with organic substances. Concurrently, the possibility of the cultivation of energy crops (requiring a greater input of nitrogen) on the AF at the plant will offer the potential to deposit larger amount of process yeast wastewater in these fields. This alternative can lead to a theoretical further increase in the yeast ($Y_p$) and fertilizer ($F_p$.) output. By contrast, the cultivation of energy plants located in the AF of the factory will lead to a decrease in the operating costs of the ES.

**Author Contributions:** Data curation, B.W.; Investigation, P.P.W. and B.W.; Methodology, P.P.W.; Supervision, P.P.W.; Analyzed the data, P.P.W. and B.W. Wrote the paper; P.P.W. and B.W.

**Funding:** This research received no external funding.

**Acknowledgments:** This work was created as a result of scientific internship of the authors in the Lesaffre Polska yeast factory (Wołczyn, Poland) and in the Poltava State Agrarian Academy (Poltava, Ukraine).

**Conflicts of Interest:** The authors declare no conflict of interest.

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
