# Peer review of "Analysis of the Potential of an Increase in Yeast Output Resulting from the Application of Additional Process Wastewater in the Evaporator Station"

_applsci, doi:10.3390/app9112282_

Round 1
Reviewer 1 Report
Authors have made acceptable modifications to the manuscripts. It now can be published in it's current form.
Author Response
First of all, we would like to thank your review, time and nice comments about this article. We really appreciate them. Thank for your suggestions in order to improve our manuscript. In addition, these comments will also be taken into account when manuscripts are compiled in the future.
Reviewer 2 Report
Generalities
The manuscript entitled “Analysis of the potential for the increase of yeast output resulting from application of additional process wastewater in the evaporator station” aims to analyze the data from a yeast factory to increase its productivity. The article is well-written, but it has a typo mistakes in all the equations that have been used for the analysis. One assumes that all equations were used correctly, but in the present form, it cannot be demonstrated.
Specifications
1) Equations 1, 1a, 2, 3, 4, 5, 6, 7, 8, 9, 10, 11, 12, 13, 14, 15, 16, 17, 18, 19, 20, 21, 22, 23, 24, 25, 26, 27 present characters that cannot be properly seen in pdf format. Please revise.
2) Line 415. This sentence is difficult to understand. Please revise.
Author Response
First of all, we would like to thank your review, time and nice comments about this article. We really appreciate them. Thank for your suggestions in order to improve our manuscript. After taking into account your suggestions in the present work, the following scope of work was realized:
- the equations was corrected (all equations were rewritten from scratch),
- the sentence difficult to understand (line 415) was corrected.
In addition, these comments will also be taken into account when manuscripts are compiled in the future.
Reviewer 3 Report
The manuscript of Włodarczyk and Włodarczyk is an analysis of the wastewater treatment in a yeast production plant. By directing the process wastewater to the evaporator station, the resulting wastewater has a reduced nitrogen concentration so that it can be used for irrigation in agriculture.
For me it took quite a while to understand the aim of the study. The main reason for this was the complicated language. I would suggest a thorough language editing.
Reducing the nitrogen in the wastewater opens the possibility to produce more yeast in the same factory. This result is not surprising; however, the complete picture is probably more complicated. Directing more wastewater to the evaporator increases the energy costs and effects the economy of the process. These things should be at least discussed.
Line 91: what are ‘carious compounds’?
Author Response
We would like to thank you for your review.
The manuscript has undergone English language editing by MDPI. The text has been checked for correct use of grammar and common technical terms, and edited to a level suitable for reporting research in a scholarly journal by native English speaking editors (MDPI Certificate).
The result of produce more yeast by reducing the nitrogen in the wastewater was known also for authors. It was the starting point for our analysis. However, our manuscript shows the possibility of achieving this assumption, and shows the path of implementation for a yeast industry. The purpose of our paper involved involves the determination of the theoretical and feasible value of production growth, based on the variation in the amount of technological wastewater designed for application in evaporator station. This analysis of wastewater streams is based on data obtained from the real yeast factory. During the three-month authors' scientific internship in the analysed factory, the problem of increasing production due to the new nitrogen directives appeared. Therefore, the authors attempted to analyse the path of reducing of the nitrogen concentration. The analysis based also on the guidelines contained in Regulation of the Minister of Environment and Directive of the European Parliament guiding the amount of substances introduced into soils and waters.
Theoretically, when a larger amount of wastewater is routed to process in an evaporation station, the possibility of using larger amounts of process wastewater in agricultural fields is feasible, which, in turn, can lead to an increase in the yeast output. Previous studies of authors have shown the necessity of a detailed analysis of the wastewater treatment system in the yeast factory. The analysed factory includes a three-stage wastewater treatment system, comprising comprised of an agricultural fields, an evaporator station, and a biological wastewater treatment plant.
Of course, as a result of the increase in the amount of process wastewater applied in to the evaporator station, the evaporator station is faced with the need to apply greater volumes of energy to operate. Directing more wastewater to the evaporator station increases the energy costs and affects the economy of the process. But, the analysed factory includes a boiler burning biomass. The share of biomass is constantly increasing. In this way, a reduction in the operating costs of the evaporator station is feasible. In addition, the cost of using an evaporator station can be compensated as a consequence of the increase in revenues resulting from the increase in yeast and fertilizer production. This information is included in the Discussion and Conclusions chapter. Moreover, the analysis of further cost reduction is actually developed jointly with Institute of Economy as the separate manuscript.
The authors re-analysed the comments of Reviewers. Next, based on data from factory the authors re-analysed the path to obtain increased production. The authors maintain their opinion that their path of analysis (including the research design, the describe of methods, the introduction and resulting from these the presentation of research results) is correct.
The ‘carious compounds’ term was corrected as the ‘humus compounds’.
Thank for your suggestions in order to improve our manuscript. In addition, these comments will also be taken into account when manuscripts are compiled in the future.

Round 2
Reviewer 3 Report
The comments of the reviewers have been addressed.
This manuscript is a resubmission of an earlier submission. The following is a list of the peer review reports and author responses from that submission.
Round 1
Reviewer 1 Report
This manuscript presented an interesting aspects of yeast factory wastewater treatment. It is a very important aspect to achieve sustainability in the industry. The analysis of wastewater management in the paper is very useful for yeast industry. There are few things should be addressed by the authors before this can be considered for publication in Applied Science.
Introduction can be improved by adding some results obtained from the analysis instead of just describing the importance and significance of the study. That will help the readers a lot.
It would be interesting if the authors can provide any case study information.
Author Response
The analysis of the reviewer feedback contributed to the improvement of our study (Analysis of production increase possibility in yeast factory based on the wastewater treatment system) in the introduction (also in improvemen of calculation and results), the description of the experiment, as well as the discussion of the results.
The authors would like to thank the Reviewer for their valuable insights which helped the authors to improve this paper. In addition, these comments will also be taken into account when manuscripts are compiled in the future.
After taking into account the suggestions of the Reviewer in the present work, the following scope of work was realized:
- language was corrected,
- the abstract and introduction was corrected,
- the methodology followed in the study was improved,
- tables of the data from analyzed factory was added (Table 1, Table 2, Table 3),
- the calculation in the study was corrected with data from factory,
- presentation of the results was improved (Figure 3),
- discussion of the results was reorganized and completed,
- the literature was supplemented.
Reviewer 2 Report
The manuscript describes just the model concept of proposed system and there are no detailed calculated results or parameters. Thus reviewer couldn’t understand the importance and novelty of the study. Moreover, quantitative data is not presented, so validity of proposed analysis system is unknown.
Author Response
The analysis of the reviewer feedback contributed to the improvement of our study (Analysis of production increase possibility in yeast factory based on the wastewater treatment system) in the introduction (also in improvemen of calculation and results), the description of the experiment, as well as the discussion of the results.
After taking into account the suggestions of the Reviewer in the present work, the following scope of work was realized:
- language was corrected,
- the abstract was corrected,
- the introduction was corrected,
- the methodology followed in the study was improved,
- tables of the data from analyzed factory was added (Table 1, Table 2, Table 3),
- the calculation in the study was corrected with data from factory,
- presentation of the results was improved (Figure 3),
- discussion of the results was reorganized and completed,
- the literature was supplemented.
The authors would like to thank the Reviewer for their valuable insights which helped the authors to improve this paper. In addition, these comments will also be taken into account when manuscripts are compiled in the future.
Reviewer 3 Report
The paper “Analysis of product increase possibility in yeast factory based on the wastewater treatment system” deals with an interesting subject. Unluckily, the paper is not well written and several statements can be discussed. Some of the drawbacks follow:
- The English is defective, until the point that several sentences cannot be understood (syntax, spelling, etc.) or are nonsense as are written
- The title does not correctly reflect the contents
- The wording is not the usual in the field
- The fate of nitrogen in the descriptions tend to be excessively simplified; i.e. the role of the different N species is not considered
- The introduction is too general and does not state the framework of the paper
- There are a number of repetitions all along the paper
- The list of streams in the lines 103 to 117 is separated from the figure
- The basic role of the microorganisms (mainly bacteria) is not considered
Round 2
Reviewer 2 Report
Basically, the revised manuscript is same for original version. This is just a case study report analyzed only the one factory. Reviewer can't think this is scientific research paper.